# Rise, and pronounced regional variation, in methylphenidate, amphetamine, and lisdexamfetamine distribution in the United States

Sneha M. Vaddadi[1], Nicholas J. Czelatka[1], Belsy D. Gutierrez[1,2], Bhumika C. Maddineni[3,4], Kenneth L. McCall[5] and Brian J. Piper[1,4]

[1] Medical Education, Geisinger Commonwealth School of Medicine, Scranton, PA, United States of America
[2] Biology, University of Scranton, Scranton, PA, United States of America
[3] Medicine, University of Texas Southwestern Medical Center, Dallas, TX, United States of America
[4] Center for Pharmacy Innovation and Outcomes, Forty Fort, PA, United States of America
[5] Pharmacy, University of New England, Portland, ME, United States of America

Corresponding author
Brian J. Piper,
bpiper@som.geisinger.edu,
psy391@gmail.com

## ABSTRACT

**Background**. The prescription stimulants methylphenidate, amphetamine, and lisdexamfetamine are sympathomimetic drugs with therapeutic use. They are designated in the United States as Schedule II substances, defined by the 1970 Controlled Substances Act as having a "high potential for abuse". Changing criteria for the diagnosis of Attention Deficit Hyperactivity Disorder in 2013 and the approval of lisdexamfetamine for binge eating disorder in 2015 may have impacted usage patterns. This report compared the pharmacoepidemiology of these stimulants in the United States from 2010–2017.

**Methods**. Distribution of amphetamine, methylphenidate, lisdexamfetamine were examined via weights extracted from the Drug Enforcement Administration's (DEA) Automated Reports and Consolidated Ordering System (ARCOS). Median stimulant Daily Dosage per patient was determined for a regional analysis. The percent of cost and prescriptions attributable to each stimulant and atomoxetine in Medicaid from the "Drug Utilization 2018 - National Total" from the Centers for Medicare and Medicaid was determined.

**Results**. There was a rise in amphetamine (+67.5%) and lisdexamfetamine (+76.7%) from 2010–2017. The change in methylphenidate (−3.0%) was modest. Persons/day stimulant usage was lower in the West than in other US regions from 2014-2017. There was a negative correlation ($r(48) = −0.43$ to $−0.65$, $p < .05$) between the percent Hispanic population per state and the Daily Dosage/population per stimulant. Methylphenidate formulations accounted for over half (51.7%) of the $3.8 billion reimbursed by Medicaid and the plurality (45.4%) of the 22.0 million prescriptions. Amphetamine was responsible for less than one-fifth (18.4%) of cost but one-third of prescriptions (33.6%). Lisdexamfetamine's cost (26.0%) exceeded prescriptions (16.3%).

**Conclusion**. The rising amphetamine and lisdexamfetamine distribution may correspond with a rise in adult ADHD diagnoses. Regional analysis indicates that stimulant

distribution in the West may be distinct from that in other regions. The lower stimulant distribution in areas with greater Hispanic populations may warrant further study.

# INTRODUCTION

Stimulants are sympathomimetic substances that mimic the effects of the sympathetic nervous system and have Food and Drug Administration-approved indications (*Coghill et al., 2014*; *King et al., 2018*). Methylphenidate, amphetamine, and other stimulants are widely used in the US for Attention Deficit Hyperactivity Disorder (ADHD), a psychiatric disorder characterized by hyperactivity, inattention, executive function deficits and emotional dysregulation (*Bădescu et al., 2016*; *Subcommittee on Attention-Deficit/Hyperactivity Disorder & Steering Committee on Quality Improvement & Management, 2011*). ADHD is one of the more common biopsychosocial disorders, with an estimated national prevalence of 9.4%, or 6.1 million, in children from age 2–17 in 2016 per the National Survey of Children's Health. Of those currently diagnosed with ADHD, over three-fifths (62.0%) were taking medication, accounting for 5.1% of the children from age 2–17 in the US (*Danielson et al., 2018*). With the implementation of the Patient Protection and Affordable Care Act in 2014 and the transition to a "fee-for-service" program in 39 states by 2016, Medicaid is the largest payer for mental health services in the US. Medicaid is a federal and state program that assists in healthcare costs for patients with limited resources, and provides insurance to over 36 million children as of June 2020, offering coverage to a sizable portion of patients on stimulant medications (*Chorniy, Currie & Sonchak, 2018*). Amphetamine-type stimulants accounted for the highest proportion of medication expenditures for Medicaid-enrolled children (*Cohen et al., 2017*). To date, the twelve states that have not elected to expand Medicaid are located primarily in the South and Midwest (*Kaiser Family Foundation, 2021*). Despite the growth in ADHD, there is variation within cultural communities. Although one of the largest ethnic minorities in the United States, Hispanic youths have reduced ADHD diagnoses and stimulant use (*Pennap et al., 2017*; *Velasco-Mondragon et al., 2016*).

ADHD diagnosis and treatment often extend beyond childhood. With a revised ADHD criterion released in 2013 in the Diagnostic and Statistical Manual (DSM) 5, ADHD diagnostic criteria has been more inclusive of adolescents and adults. As of 2018, approximately 4% of US adults are afflicted with ADHD and two-thirds of children continue to experience at least one ADHD symptom throughout their lives (*Wei et al., 2018*). Lisdexamfetamine and mixed amphetamine salts were found to cause a significant improvement in adult ADHD symptoms without symptoms rebound after ceasing medication (*Buoli, Serati & Cahn, 2016*; *Stevens, Wilens & Stern, 2013*). In contrast, two Cochrane reviews concluded the quality of evidence base assessing the efficacy of amphetamine for child, adolescent, and adult ADHD was "low to very low"

(*Castells, Blanco-Silvente & Cunill, 2018*; *Punja et al., 2016*). Stimulants have also been approved for other medical conditions, such as the approval of lisdexamfetamine for binge eating disorder in 2015 (*Guerdkikova et al., 2016*).

Stimulant use has been associated with mild adverse effects such as appetite and sleep disturbances that impact quality of life, modest reductions in height in children and adolescents, and small elevations in blood pressure and heart rate but the long-term adverse effects of these substances are not well established (*Coghill et al., 2014*; *Groenman et al., 2017*). A recent study determined that patients prescribed stimulants had a nine-fold elevated risk of developing basal ganglia and cerebellar disorders, and it is suggested by the authors that this may be indicative of the ADHD phenotype as increased risk of these motor disorders (*Curtin et al., 2018*). Interestingly, a history of recreational methamphetamine or amphetamine misuse was associated with a three-fold elevated risk of developing Parkinson's (*Curtin et al., 2015*). A meta-analysis of cross-sectional Positron Emission Tomography investigations showed that long-term blockade of the dopamine transporter with ADHD pharmacotherapies caused neuroadaptive striatal elevations in this protein (*Fusar-Poli et al., 2012*) which was subsequently confirmed in a longitudinal report (*Wang et al., 2016*).

The stimulants methylphenidate, amphetamine, and lisdexamfetamine are classified by the US Drug Enforcement Administration (DEA) as Schedule II drugs, which are defined as those with a ''high potential for abuse, with use potentially leading to severe psychological or physical dependence'' (*DEA, 2021*). Methylphenidate had a slightly higher affinity for the dopamine and norepinephrine transporters than cocaine (*Han & Gu, 2006*). Methylphenidate, D-amphetamine, and cocaine usually share discriminative stimulus effects (*Kollins, MacDonald & Rush, 2001*). The prodrug lisdexamfetamine may be resistant to misuse but that does not prevent consuming multiple doses orally. Several data sources have produced evidence indicating there is appreciable non-medical use of prescription stimulants. The Monitoring the Future survey of recreational drug use determined that 4.6% of 12th graders misused Adderall (amphetamine) in 2018 (*National Institute on Drug Abuse et al., 2018*). Five million people misused a prescription stimulant in the US in the past year (*Substance Abuse and Mental Health Services Administration , 2019*). Use and misuse may be particularly high among some populations.

In one institution in Puerto Rico, half of medical students reported a history with prescription stimulants and 89.4% of this subset used these agents without a prescription (*Acosta et al., 2019*). Calls to US poison control centers between 2007 and 2012 ($N = 23,533$) were more likely to involve amphetamine than the pro-drug lisdexamfetamine (*Kandland & Klein-Schwartz, 2015*). Exposures involving ADHD medications ($N = 156,635$, *i.e.,* one call to poison control every fifty minutes) for patients age $\leq 19$ increased by 71.2% from 2000 to 2011 (*King et al., 2018*). The number of population corrected exposures was 42.0% lower in the western relative to the midwestern states (*King et al., 2018*).

With increasing prevalence of medicated ADHD in children and adults as well as lisdexamfetamine in adults with binge eating disorder, there is a greater need to understand the extent of stimulant use nationally. This report utilized the US DEA's Automated Reports and Consolidated Ordering Systems (ARCOS) comprehensive database to evaluate changes

in the use of amphetamine, methylphenidate, and lisdexamfetamine nationally from prior years to 2017. We extended upon past research (*Piper et al., 2018a*) by investigating the overall change of stimulant use from 2010–2017. We then calculated Daily Dosage values to investigate the change in use from 2016 and 2017. We also explored variations in use in the Hispanic population and geographical regions (*King et al., 2018*). Finally, we explored stimulant use and expenditures within Medicaid via Medicaid reimbursements to healthcare facilities.

## MATERIALS & METHODS

### Data sources

Stimulant data was extracted from the DEA's ARCOS, a national database containing a yearly updated report of retail drug distribution from manufacturers and distributors (*Drug Enforcement Administration, 2017*). Extracted data included total grams of stimulant use (two-hundred 5 mg pills = 1 g) per drug per state (50 states excluding US territories) from 2010 to 2017. Three Schedule II stimulants were examined: amphetamine, methylphenidate and lisdexamfetamine. This database has been frequently used in prior pharmacoepidemiology reports (*Atluri, Sudarshan & Manchikanti, 2014*; *Bokhari, Mayes & Scheffler, 2005*; *Collins et al., 2019*; *Davis et al., 2020*; *Pashmineh Azar et al., 2020*; *Piper et al., 2020*; *Simpson et al., 2019*). The main unit of data reported of drug by weight may be less familiar than other units like the number of prescriptions. ARCOS was validated by examining the total weight of oxycodone in this database relative to that reported in a Prescription Monitoring Program which revealed a high correlation ($r = 0.99$) (*Piper et al., 2018b*). Further, a comparison of stimulant use by weight by zip code in a high versus low classification indicated an excellent concordance (96.5%) between ARCOS and California's Prescription Drug Monitoring Program (*Bokhari, Mayes & Scheffler, 2005*).

Our goal was to examine the change in stimulant use and in number of patients utilizing Schedule II stimulants, but ARCOS data is limited to the total quantity of Schedule II stimulants distributed in a geographical location. To approximate the changes in number of patients utilizing these substances, we calculated the median estimated Daily Dosage per person (mg/person/day) for each stimulant. These values were calculated from 2018 de-identified data ($N = 88,202$) from the electronic health record (EHR) of the Geisinger Health System, an integrated health delivery system in central and northeastern Pennsylvania. The calculated values, termed "Daily Dosage", are 20 mg/day/person for methylphenidate and amphetamine and 40 mg/day/person for lisdexamfetamine based on the median dose prescribed to patients. These values were then used for determination of the change in Daily Dosage in 2016–2017 and regional comparison by dividing total grams extracted from ARCOS by Daily Dosage data (*e.g.*, methylphenidate = 20 mg). An analysis was completed with state-specific Hispanic population data. The percent Hispanic population per state was obtained from the demographic profiles from the Pew Research Center (*Pew Research Center, 2021*). Trends of stimulant use in the Medicaid system were investigated using data from the "Drug Utilization 2018 - National Total" from the Centers for Medicare and Medicaid (*Centers for Medicare and Medicaid Services, 2021*).

This form was used to extract the number of prescriptions and total cost of substances that were reimbursed to prescribing healthcare facilities by Medicaid for methylphenidate, lisdexamfetamine, and amphetamine. Atomoxetine (non-scheduled and therefore not reported in ARCOS) was also obtained. IBM Micromedex was used to obtain formulation names. Institutional Review Board approval was obtained from the University of New England (#20180410-009) and Geisinger (2019-0598).

## Data analysis

Total stimulant use from 2010–2017 was investigated by averaging extracted stimulant weight (grams) per state and comparing per year for each stimulant. Data was deemed to be significant ($p < 0.05$) after a paired $t$-test. This provided an overall index of the temporal profile. For further investigation into change from 2016–2017, ARCOS aggregate data was divided by median Daily Dosage values (mg/person/day). These values were divided by population per state and labeled "Daily Dosage/population". Conversion to Daily Dosages provides units that some may find more intuitive than total weights. The percent change in these values per state per drug from 2016–2017 was calculated and compared to provide recent changes and extend upon our prior report (*Piper et al., 2018a*; *Piper et al., 2018b*). Values were deemed significant as ≥1.96 standard deviations above and below the mean, accounting for values outside a 95% confidence interval. A heat map was constructed with Excel using the percent data detailed above and with the values from 2017. Regional variance analysis was conducted by dividing ARCOS aggregate data per US region (Midwest, Northeast, South, West) and dividing by Daily Dosage. These values were labeled "Person/day" and compared for all three stimulants from 2014–2017. Data was deemed to be significant ($p < 0.05$) after an unpaired $t$-test. For all above calculations, outliers were determined through a Grubbs analysis and significant values were excluded. A correlational analysis was done with Daily Dosage/population values and Hispanic population data per state in 2016 and 2017. This was completed to verify the utility of the Daily Dosage measure and to extend upon earlier findings (*Piper et al., 2018a*). The amount of Medicaid spending ($) and number of prescriptions in 2018 for each Schedule II stimulant and atomoxetine were extracted and each presented as percentage of a whole. Health care spending in the US was 17.7% of the nation's gross domestic product (*Centers for Medicare and Medicaid Services, 2021*) so characterizing sources of spending for common pharmacotherapies is an important concern. Identifying the significance of spending in a major public program such as Medicaid, utilized by millions of children in the United States, will further justify the need for investigation in stimulant use. Alpha values obtained that were below the standard ($p < .05$) threshold were noted. Variance was reported as the standard error of the mean (SEM).

## RESULTS

Stimulant use increased +67.5% for amphetamine and +76.7% for lisdexamfetamine from 2010-2017 on average across all fifty states. In contrast, methylphenidate use decreased slightly (−3.0%). For amphetamine and lisdexamfetamine, there was a significant increase in the total stimulant use compared to 2010 starting from 2014 (Fig. 1A). Further

investigation into the change from 2016–2017 was completed with a daily dose and population-corrected analysis. The percent change in Daily Dosage/population across fifty states (with New Mexico excluded as an outlier) was +4.6% for amphetamine, +2.3% for lisdexamfetamine, and −1.4% for methylphenidate (Figs. 1B, 1C). The preponderance (85.0%) of states increased their amphetamine and over two-thirds (72.0%) increased their lisdexamfetamine use. In contrast, 86.0% of states decreased their methylphenidate use. Wisconsin, South Dakota and West Virginia were all significantly lower than the mean for amphetamine. Hawaii had a significantly lower value while Nevada and South Dakota had greater values compared to the national average for methylphenidate. For lisdexamfetamine, South Dakota had a significantly lower value while Wisconsin had a greater value relative to the mean (Fig. 1D).

The Daily Dosage/population values for 2017 are depicted in the heat map in Fig. 2A, indicating pronounced regional variance. There was a six-fold difference between values for the highest (12.2) and lowest (2.9) states. Six states with the lowest values of Daily Dosage/population were all in the Western region of the United States. A regional analysis was completed to further investigate geographic-based variability. A comparison of persons/day per region from 2014–2017 revealed a significant difference in the values for the Midwest, Northeast, and South compared to the West consistently from 2014–2017 (Fig. 2B). Analysis with the percent Hispanic population and Daily Dosage/population identified a negative correlation for each stimulant in 2016. These findings were replicated for 2017 (Fig. 3). States with a greater portion of Hispanic populations had lower stimulant distributions.

The Medicaid program spent $3,765,776,679 for 22,024,008 prescriptions for methylphenidate, lisdexamfetamine, amphetamine and atomoxetine in 2018. Methylphenidate accounted for about half of cost (51.7%) and prescriptions (45.4%). Amphetamine was responsible for less cost (18.4%) than prescriptions (33.6%). Lisdexamfetamine's cost (26.0%) exceeded prescriptions (16.3%). Further analysis was completed examining the breakdown of different formulations (Table 1).

## DISCUSSION

This study investigated the trends in Schedule II stimulant use in the United States from prior years to 2017, and indicated an overall increase. An analysis for total grams from 2010–2017 and a Daily Dosage and population-considered analysis from 2016–2017 indicated a rise in amphetamine and lisdexamfetamine and no appreciable change in methylphenidate.

This pattern of stimulant use change may be at least partially explained by an increase in adult ADHD diagnoses and treatment (*Bădescu et al., 2016*). The revision of the ADHD criteria in the DSM-5 was more inclusive of adult ADHD, leading to more adults meeting the requirements for diagnosis than that for the DSM-IV (*Epstein & Loren, 2013*). A 2018 study looking at ADHD treatment in privately-insured women aged 15–44 found a similar pattern in these three stimulants as this study, with the largest change in stimulant use being in the age range of 25–29 years (*Anderson et al., 2018*). Studies have also indicated variance

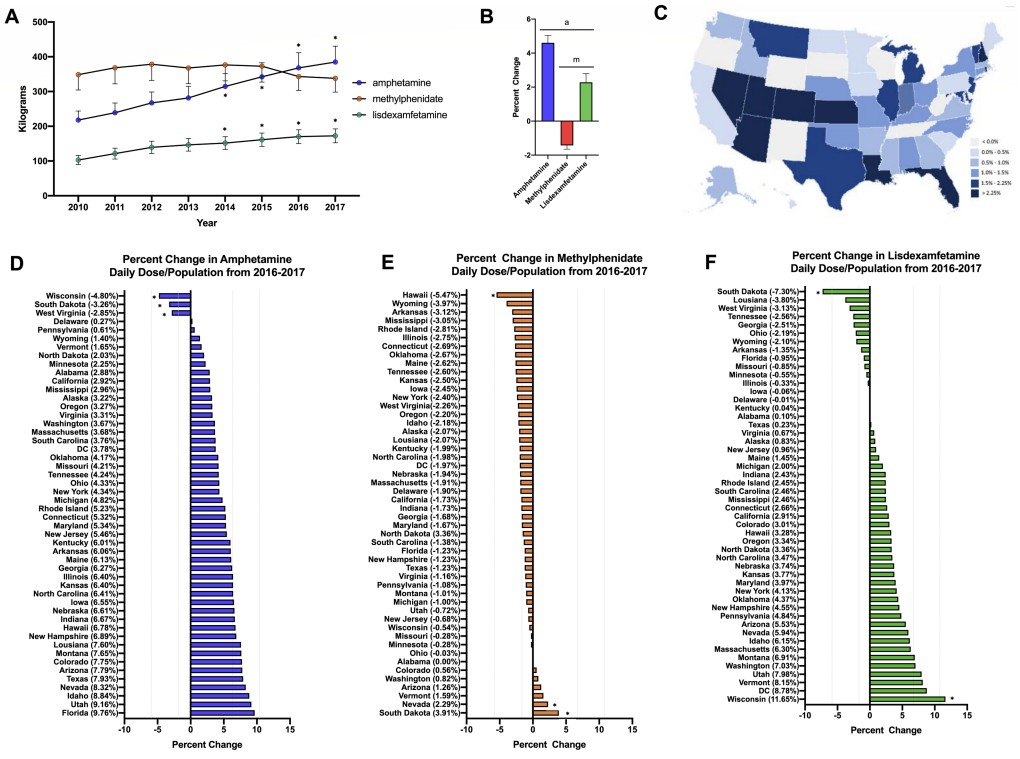

**Figure 1** **Weight per stimulant per state.** (A) Weight per stimulant per state showed a 67.5% and 76.7% increase in amphetamine (*paired $t$-test $p < 0.05$ in comparison to 2010) and lisdexamfetamine (*paired $t$-test $p < 0.05$ in comparison to 2010) and a 3.0% decrease in methylphenidate (paired $t$-test $p = 0.8590$). (B) Average percent change of 50 states data in Daily Dosage/Person from 2016-2017 for amphetamine (+4.6%), lisdexamfetamine (+2.3%), and methylphenidate (−1.4%). Lisdexamfetamine was significantly different from amphetamine ($^a t$-test $p < 0.05$) and methylphenidate ($^m$t-test $p < 0.05$). (C) Heat map of United States depicting percent change in total stimulant Daily Dosage/population per state from 2016–2017. (D–F). Percent change in Daily Dosage and population-adjusted analysis per state for amphetamine (D), methylphenidate (E), and lisdexamfetamine (F) reveals 85.0% of states increased their amphetamine, 72.0% increased their lisdexamfetamine, and 86.0% states decreased their methylphenidate use. Significant states were marked if 1.96*SD greater or less than the mean for each stimulant. The percent change for New Mexico was excluded as an outlier (−17.45% for amphetamine, −19.98% for methylphenidate, and −36.63% for lisdexamfetamine).

in drug efficacy for long-term treatment in adult ADHD. Although methylphenidate is considered a first-line treatment for child and adolescent ADHD, the long-term (>12 months) efficacy of stimulants is not well established (*Punja et al., 2016*). Alternate uses for stimulants outside of AHDH treatment may also contribute to these patterns in stimulant use. Lisdexamfetamine is a well-tolerated treatment for moderate to severe binge eating disorder (*Heo & Duggan, 2017*). Amphetamine and methylphenidate are also employed in the treatment of apathy in Alzheimer's patients and other neuropsychiatric conditions in the elderly (*Dolder, Davis & McKinsey, 2010*). Further exploration is needed on how the expansion of stimulant use in other neuropsychiatric conditions like binge eating disorder (*Heo & Duggan, 2017*) impacts the use of stimulants.
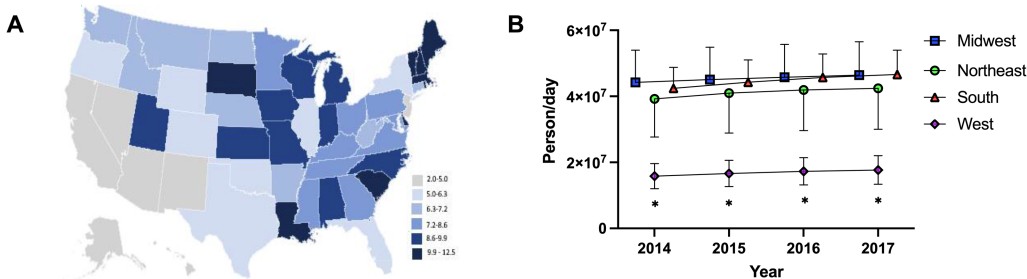

**Figure 2** **Heat map and regional analysis.** (A) Heat map for Daily Dosage/Population per state for 2017. (B) Person /Day per Region from 2014–2017 indicated a significant difference with the West compared to the South, Midwest and Northeast from 2014–2017 (* $t$-test $p < 0.05$). Time points were slightly offset for display purposes.

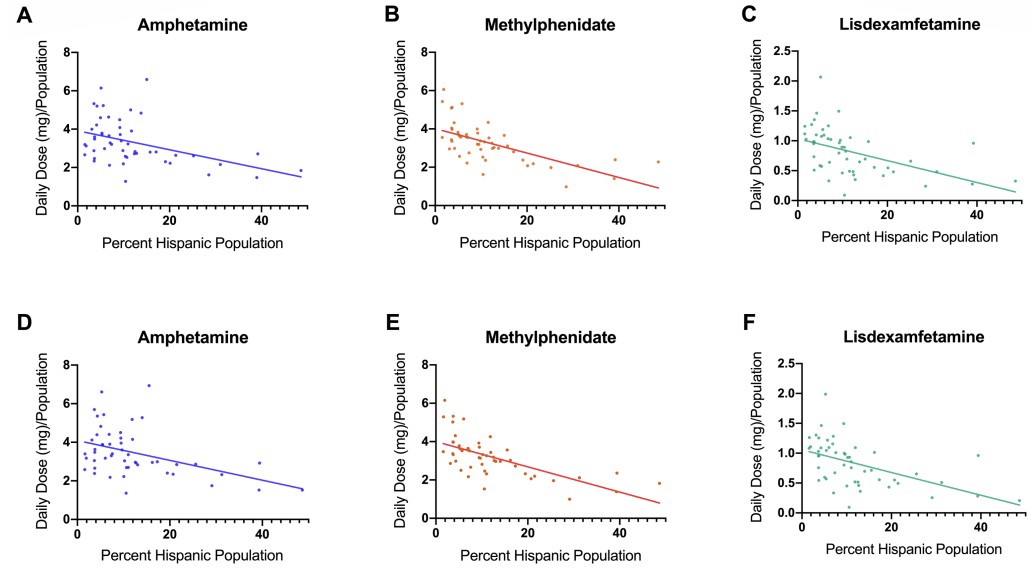

**Figure 3** **Scatterplots and negative correlations.** Negative correlations between total Daily Dosage/Population value and the percent Hispanic per state for 2016. (A, amphetamine: $r$ (48) = −0.43, $p = 0.0017$; B, methylphenidate: $r$ (48) = −0.64, $p < 0.0001$; C, lisdexamfetamine: $r$ (48) = −0.49, $p < 0.0001$). Negative correlations between total Daily Dosage/Population value and percent Hispanic per state for 2017 ((D) amphetamine $r$ (48) = −0.43, $p < 0.005$; (E), methylphenidate: $r$ (48) = −0.65, $p < 0.0001$; (F), lisdexamfetamine: $r$ (48) = −0.52, $p < 0.0001$).

The percent Hispanic population had a negative correlation with stimulant use per state for 2016 and 2017. Other studies have also indicated a lower stimulant use by Hispanic children compared to their non-Hispanic peers (*Davis et al., 2019*). Young Hispanic adults and children have a significantly lower use of outpatient mental health services for mental health and substance abuse care (*Marrast, Himmelstein & Woolhandler, 2016*). This correlation of lower stimulant use in states with greater Hispanic population may indicate a lower use of stimulants among the Hispanic population. This may be attributed to difficulties with access to healthcare, as prior to the implementation of the Affordable

**Table 1** Percent of Medicaid amount reimbursed and number of prescriptions per stimulant (A) and by formulation for methylphenidate (B), amphetamine (C) and atomoxetine (D).

| | % Amount reimbursed | % Number of prescriptions |
|---|---|---|
| **A. Total** | | |
| methylphenidate | 51.67 | 45.36 |
| lisdexamfetamine | 25.97 | 33.62 |
| amphetamine | 18.42 | 16.27 |
| atomoxetine | 3.95 | 4.75 |
| **B. Methylphenidate** | | |
| methylphenidate | 48.00 | 62.51 |
| Focalin | 25.81 | 14.16 |
| Concerta | 11.61 | 6.11 |
| dexmethylphenidate | 6.39 | 12.18 |
| Quillichew | 3.26 | 1.86 |
| Quillivant | 1.45 | 0.79 |
| Aptensio X | 1.13 | 0.91 |
| Daytrana | 1.05 | 0.62 |
| Cotempla X | 0.83 | <.01 |
| other | 0.45 | <.01 |
| **C. Amphetamine** | | |
| Adderall | 51.46 | 21.54 |
| mixed amphetamine | 20.89 | 24.62 |
| dextroamphetamine | 20.32 | 51.59 |
| Adzenys | 2.52 | 0.77 |
| Dyanavel X | 2.44 | 0.79 |
| Mydayis | 0.90 | <.01 |
| Evekeo | 0.67 | <.01 |
| Procentra | 0.50 | <.01 |
| Other | 0.30 | <.01 |
| **D. Atomoxetine** | | |
| atomoxetine | 75.17 | 91.49 |
| Strattera | 24.83 | 8.50 |

Care Act (ACA) in 2014, 30% of Hispanics reported no health insurance compared to 11% of non-Hispanic whites (*Velasco-Mondragon et al., 2016*). Along with social factors such as language barriers, cultural factors such as a perceived difference in the need for outpatient mental health care may also explain differences in resource utilization (*Alergria et al., 2002*). The cumulative effect of these sociocultural or pharmacoeconomic factors may lead to individuals being unable or hesitant to seek medical attention for ADHD symptoms.

Our regional analysis with data controlled for Daily Dosage found that the West has a significantly lower Schedule II stimulant use compared to the South, Northeast, and Midwest. This pattern was seen in other studies spanning from 1998–2018 focusing on both child and adult ADHD, with the West having the lowest ADHD prevalence or change in stimulant use (*Huber et al., 2018*; *Piper et al., 2018a*; *Xu et al., 2018*). The availability of

specialty health care providers in the US exhibits pronounced regional differences. One study indicated a nine-fold difference of psychiatrists per one-hundred thousand state population between the highest and lowest states (*Beck et al., 2018*). The West contained half of the ten states with the fewest psychiatrists while the Northeast area contained nine of ten states with the most psychiatrists (*Beck et al., 2018*). An earlier ARCOS report determined that counties with greater stimulant use had more physicians per capita and were more affluent (*Bokhari, Mayes & Scheffler, 2005*). Calls to poison control centers involving ADHD medications were lower in the West relative to the South and Midwest (*King et al., 2018*). Though frequently reported, this pattern has little explanation and may be due to various factors. A 2015 report suggests that with many states of higher altitude located primarily in the West, the altitude may serve as a protective factor against ADHD by increasing dopamine levels (*Huber et al., 2018*). However, others are skeptical of the altitude hypothesis for lower ADHD levels in the Rocky Mountain states (*Arns, Swanson & Arnold, 2018*). As noted above, cultural diversity may also play a role. Of the youth in California, almost 40% are Hispanic, an ethnicity that has significantly lower stimulant use (*Pennap et al., 2017*). Other factors may also contribute to these regional differences including pharmacy policies, Medicaid policies, rates of uninsured, or state laws regarding use of psychiatric medications (*Fulton, Scheffler & Hinshaw, 2015*).

It was also noteworthy that secondary analyses within Medicaid identified much greater use of methylphenidate than lisdexamfetamine and other stimulants which was unlike that observed with ARCOS. Prior Medicaid research has examined stimulants as a group without differentiation of individual agents (*Cummings et al., 2017*; *Ji et al., 2018*; *Raghavan et al., 2012*). It is notable that twenty-seven states in 2015 had enacted Medicaid policies to be congruent with the 2011 American Academy of Pediatrics guidelines that clinicians refer parents of preschoolers (age 4–5) for training in non-pharmacological behavior therapy and subsequently only treat with medication if the behavioral treatment failed to sufficiently improve functioning (*Wolraich et al., 2011*). Seven states (AZ, FL, IL, LA, MA, VA, and WV) required use of non-medication before medication (*Hulkower et al., 2017*). The pattern of prescriptions, and expenditures by formulation among Medicaid recipients warrants further attention in updates of this study.

Despite stimulants being classified as Schedule II substances due to their potential for abuse, national surveys of adolescents and young adults indicate that a subset of prescribed stimulants are used for non-medical purposes (NIDA, 2019) or may result in calls to poison control centers (*King et al., 2018*). In addition to misuse potential, another concern with the expanding use is the potential of drug-drug interactions with many medications, including MAO oxidase inhibitors, vasopressors, and coumadin anticoagulants (*Groenman et al., 2017*; *Subcommittee on Attention-Deficit/Hyperactivity Disorder & Steering Committee on Quality Improvement & Management, 2011*). Based on total amount reimbursed, Medicaid data indicates that approximately 50% of the expense is due to methylphenidate use, while lisdexamfetamine expense was approximately 25%. Stimulants accounted for over one-fifth of the outpatient medication expenditure by Medicaid for children (*Cohen et al., 2017*) and supports the need for further investigation into stimulant use. Ongoing discussions among the one-fifth of the US states that have not yet expanded Medicaid (*e.g.*, Georgia, Florida,

North Carolina, Wisconsin) (*Kaiser Family Foundation, 2021*) may encourage increased attention on pharmacoeconomics including coverage for cognitive enhancing agents like atomoxetine with the least misuse potential. Stimulant use in ADHD treatment is generally regarded as safe and efficacious when used as directed with well over a half-century history (*Coghill et al., 2014*; *Cortese et al., 2018*; *Rasmussen, 2015*). However, some studies indicate controversial safety and efficacy of these Schedule II stimulants, which is of concern with the preponderance of methylphenidate use and the increase in lisdexamfetamine (*Castells, Blanco-Silvente & Cunill, 2018*; *Cortese et al., 2018*; *Punja et al., 2016*; *Storebøet al., 2015*). This increasingly ubiquitous stimulant distribution and use indicates the need for further investigation.

There are several limitations to this pharmacoepidemiological study. Although ARCOS is inclusive of all Schedule II stimulant use, and Medicaid covers an important subset of the US population, one limitation is the use of percent Hispanic population per state as determined by the US Census for correlations with stimulant use. Analysis via zip code (*Bokhari, Mayes & Scheffler, 2005*) or a more focused geographical delineation or using EMR may provide more conclusive associations. Another caveat is the use of Medicaid reimbursement for depicting stimulant trends within the expansive population covered under this public program. The predominance of certain stimulants may be impacted by variations in what formulations are reimbursed rather than reflecting the true demand for each stimulant. Future investigations could examine the contribution of Medicaid policies or include other data sources to characterize the populations with the most pronounced changes in stimulants (*i.e.*, adults with binge eating disorder versus ADHD, preschoolers, pregnant women, or dementia patients).

## CONCLUSIONS

In conclusion, this report identified increases in distribution in amphetamine and lisdexamfetamine in the United States using DEA data. Examination of Medicaid revealed $3.8 billion USD in spending for stimulant medications in 2018. Further investigation is needed to better understand the sociocultural or economic factors mediating the pronounced regional and cultural variance observed. Increased pharmacoeconomic investigations may be warranted for this ubiquitous class of medications.

## ACKNOWLEDGEMENTS

The generation of heat maps was done with the help of Daniel Kaufman, MS. This project was also completed with the technical assistance of Iris Johnston.

### Funding

Software employed in this project was provided by the NIEHS (T32-ES007060-31A1). Sneha M. Vaddadi and Brian J. Piper were supported by the Fahs Beck Fund for Research and Experimentation. Nicholas J. Czelatka and Belsy D. Gutierrez were supported by

the Health Resources Services Administration (D34HP31025). Publication costs were generously provided by the Geisinger Commonwealth School of Medicine. The funders had no role in study design, data collection and analysis, decision to publish, or preparation of the manuscript.

## Grant Disclosures
The following grant information was disclosed by the authors:
NIEHS: T32-ES007060-31A1.
Fahs Beck Fund for Research and Experimentation.
Health Resources Services Administration: D34HP31025.
Geisinger Commonwealth School of Medicine.

## Competing Interests
Brian J Piper is part of an osteoarthritis research team supported by Pfizer. The other authors declare that they have no competing interests.

## Author Contributions

- Sneha M. Vaddadi conceived and designed the experiments, performed the experiments, analyzed the data, prepared figures and/or tables, authored or reviewed drafts of the paper, and approved the final draft.
- Nicholas J. Czelatka performed the experiments, analyzed the data, prepared figures and/or tables, authored or reviewed drafts of the paper, and approved the final draft.
- Belsy D. Gutierrez performed the experiments, prepared figures and/or tables, authored or reviewed drafts of the paper, and approved the final draft.
- Bhumika C. Maddineni performed the experiments, authored or reviewed drafts of the paper, and approved the final draft.
- Kenneth L. McCall conceived and designed the experiments, authored or reviewed drafts of the paper, and approved the final draft.
- Brian J. Piper conceived and designed the experiments, analyzed the data, prepared figures and/or tables, authored or reviewed drafts of the paper, and approved the final draft.

## Human Ethics
The following information was supplied relating to ethical approvals (i.e., approving body and any reference numbers):

Institutional Review Board approval was obtained from the University of New England (#20180410-009) and Geisinger (2019-0598).

## Data Availability
The Drug Enforcement Administration data is available in the Supplementary File.

The Medicaid data is available at the Medicaid site: https://www.medicaid.gov/medicaid/prescription-drugs/state-drug-utilization-data/index.html.

## Supplemental Information

Supplemental information for this article can be found online at http://dx.doi.org/10.7717/peerj.12619#supplemental-information.

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
