# Peer review of "Rise, and pronounced regional variation, in methylphenidate, amphetamine, and lisdexamfetamine distribution in the United States"

_PeerJ, doi:10.7717/peerj.12619_

## Round 0.1 · original submission · Major Revisions

I apologise for the time taken to obtain thorough reviews for this manuscript, I understand that it would have been frustrating for you, but it proved difficult capturing people's attention over the past few months. Both Reviewers were appreciative of this study, but I feel that the revisions suggested by Reviewer 2 would significantly increase the likely impact of this paper, and the elaboration on the Methodology would increase confidence in the analysis.

Reviewer 1 ·

Basic reporting

The article is interesting and informative.

Experimental design

Methods appear sound.

Validity of the findings

Data supports findings.

Additional comments

The manuscript is interesting and informative.

Lines 93-95 "Patients prescribed stimulants had a nine-fold elevated risk of developing basal ganglia and cerebellar disorders." This should be clarified. Curtin et al noted in their abstract: "The association of ADHD patients prescribed psychostimulants with higher risk of diseases of the basal ganglia and cerebellum may reflect a more severe ADHD phenotype rather than a direct association between prescribed stimulant use and basal ganglia or cerebellum disorders." They noted patients with ADHD had a 2.4 fold increased risk.

In the discussion, speculation of "independent spirit of the western U.S." should be backed up with a reference or removed. Much of the lower use in the may be driven by lack of specialty health care providers in several of the western states.

Line 274-277 should be rewritten as stimulants are one of the most effective classes of drugs in medicine with more than a 50 year history of safety and the comment may give the reader the false notion that these medications are not safe and effective when prescribed and used appropriately.

Reviewer 2 ·

Basic reporting

The article is interesting however it is currently not written for an international audience and could be improved. Some examples including referring to Schedule II stimulants in the title. This was also in the introduction with no definition or background (line 100), a definition was provided in the discussion, this should be removed and placed in the background. The abstract and article should be reviewed to make sure the context is understandable to an international audience including the title.
Whilst the information in the background is informative, the statistics and references are over 10 years old, for example, in line 71 you have the growth until 2011, however have not reported on any statistics since then. Please provide further references. There is a reference missing on line 87.
Line 100 you state “Non-medical use of these Schedule II agents is appreciable.” Please expand on what you mean by this.
In some instances, for example, line 105 you use brand names, it is important to be consistent and also state the drug name if it is necessary to use the brand name.
There are some occasions when the acronym is not provided in the manuscript for example line 144 electric health records and line 87 & 218 DSM-5, line 219 DSM-IV, line 280 DEA.
Rewrite line 263, ‘not insignificant portion’, it is unclear what you are trying to state here.

Experimental design

Whilst I get the sense the authors have conducted a significant amount of analysis, I am having difficulties following what was conducted and why. The article would be significantly improved if this section were rewritten in particularly the data analysis section.
Areas to help with clarification are providing context to ‘stimulant weight’ what is this referring to, breaking down your different studies into parts, explanation on how the daily doses were calculated, justification on why you choose 1.96 standard deviations from the mean. Expand on this new methodology by providing more tangible information.
You mention Drug Enforcement Administration Weights in the abstract however this is not discussed in the materials and methods section. This should be included in this section.
You state that you conducted linear regression analysis for the Hispanic population however you report on correlations, where is the linear regression model? I would recommend reviewing the statistics choices for better test statistics other than a correlation or justify why this was chosen. Further statistical investigation is warranted.

Validity of the findings

A clear description of your methods would help in understanding the results sections for example line 181 ‘stimulant use, by weight’.
Adding the p values, 95% confidence intervals and the tests conducted to all the figures would be beneficial for the reader. Whilst the figures are good, tables are also helpful to the reader, I would consider using some tables rather than all figures as this would be more informative to the readers and would provide a more in-depth understanding of the findings.
For Figure 4 and 5, it is unclear on the rationale of providing the figures and what the differences mean. Again, making it clearer to an international audience the differences between the different reimbursement, and clear about the Medicaid by cost and Medicaid by prescription. In some instances, you have overreached on your findings for example with the Hispanic populations, the methodology was a correlation, and then on line 236, you state the ‘low rate of stimulant use among the Hispanic communities’, did you also calculate rates?
It is good practice to identify your limitations in your study design, there is currently no limitation. Please reflect on your study design and add in the limitations to your study.

Additional comments

I commend the authors for the time they have taken in exploring this interesting topic and dataset. Whilst the manuscript is interesting and relevant, the manuscript should be tailored for an international audience with greater attention to details (for example with the use of acronyms), additionally, the method and result sections including the statistical methods should be improved upon before acceptance.

---

## Round 0.2 · Minor Revisions

The reviewer has noted some clarifications that still require attention. Please respond to these as appropriate.

Reviewer 2 ·

Basic reporting

Thank you to the authors for conducting a significant review of the article and incorporating the suggestions and providing adequate comments. The paper has been significantly improved as a result and is now more understandable to an international audience.
I have a couple of pending suggestions for consideration outlined below.
Non-tracked change copy:
Line 122: There is no reference to support this statement. Please add a reference.
Line 200: Please clarify this sentence, it seems conflicting. ‘Alpha values obtained that were below the standard (p < .05) threshold were noted (p < .001).’
Line 201: please write SEM in full for the first time.
Line 263: ‘..difficulties to access health care’. Can you please include a reference.
Line 263: ACA. Please write acronym in full.
Lines 275-277: Please provide a reference. The sentence refers to a study but no reference is provided.
Line 292: starting with ‘Schedule II substances….’ Removed as included in the introduction.

Figures:
Whilst I agree the figures are good and significant work has gone into producing them. My suggestion was in relation to replacing some of the figure with tables, not duplicating the figures with tables. In the epidemiology world a combination of figure and tables is sometimes more suited to the readers. For example, some clinicians prefer data tables, perceiving numbers as less “biased” than a graphic. Additionally, some figures lack specific number for example 1a, if I were to replicate your study in a different setting, I would not know what numbers to reference to enable a comparison of findings.
Figure 1:
- 1a: you state that a paired t-test was conducted however in the methods it states an unpaired t-test, could you please verify and correct the test used.
- 1a: Can you have at the start of each sentence the corresponding graph or consistently at the end of each sentence please. For example (A,…) is in the middle of the sentence.
- 1a: Was the paired t-test only done for lisdexamfetamine? In the results it has there was a significant difference also for amphetamine however the results weren’t included anywhere. Please provide the p value for all the t-tests.
- ‘Lisdexamfetamine was significantly different from amphetamine (at-test p <0.05) and methylphenidate (mp < 0.05).’ – what figure does this belong too? If this is the same as the following ‘For amphetamine and lisdexamfetamine, there was a significant increase in the
208 total stimulant use compared to 2010 starting from 2014 (Figure 1A).’ in the results section, I would remove from the figure description as it is causing confusion or add extra clarity.

Figure 4 and 5:
- I am still not sure of what the value is of the figures 4 and 5, whilst the figures are good to know, I am not sure what the rationale is in including them, you state in the methods that this information was going to be used to ‘identifying sources of potential inefficiency in spending’, however I cannot see where this is completed and how you may draw a conclusion on this. Rationale on why this is included is needed, additionally why 2018 data was selected as opposed to 2017 data to align with your study data. Having a clearer aim in relation to the use of the data and reworking the methods, will supporting in addressing this gap.
An additional review and refinement of the results and figures to ensure they align and also align with the methods would further strengthen the manuscript and improve the readability.

Experimental design

See above

Validity of the findings

See above

Additional comments

See above

---

## Round 0.3 · accepted · Accept

Thank you for addressing the Reviewers comments in your revision. I hope that things are looking up for your group in 2022.